# Differential Epigenetic Marks Are Associated with Apospory Expressivity in Diploid Hybrids of *Paspalum rufum*

**DOI:** 10.3390/plants10040793

**Published:** 2021-04-17

**Authors:** Mariano Soliman, Maricel Podio, Gianpiero Marconi, Marco Di Marsico, Juan Pablo A. Ortiz, Emidio Albertini, Luciana Delgado

**Affiliations:** 1CONICET-UNR/Laboratorio de Biología Molecular, Facultad de Ciencias Agrarias, Instituto de Investigaciones en Ciencias Agrarias de Rosario (IICAR), Universidad Nacional de Rosario, Zavalla S2123, Argentina; marianosoliman@gmail.com (M.S.); podio@iicar-conicet.gob.ar (M.P.); ortiz@iicar-conicet.gob.ar (J.P.A.O.); 2Department Agricultural, Food and Environmental Sciences, University of Perugia, 06121 Perugia, Italy; gianpiero.marconi@unipg.it (G.M.); marco.dimarsico@studenti.unipg.it (M.D.M.)

**Keywords:** apomixis, epigenetic, diploid level, *Paspalum rufum*

## Abstract

Apomixis seems to emerge from the deregulation of preexisting genes involved in sexuality by genetic and/or epigenetic mechanisms. The trait is associated with polyploidy, but diploid individuals of *Paspalum rufum* can form aposporous embryo sacs and develop clonal seeds. Moreover, diploid hybrid families presented a wide apospory expressivity variation. To locate methylation changes associated with apomixis expressivity, we compare relative DNA methylation levels, at CG, CHG, and CHH contexts, between full-sib *P. rufum* diploid genotypes presenting differential apospory expressivity. The survey was performed using a methylation content-sensitive enzyme ddRAD (MCSeEd) strategy on samples at premeiosis/meiosis and postmeiosis stages. Based on the relative methylation level, principal component analysis and heatmaps, clearly discriminate samples with contrasting apospory expressivity. Differential methylated contigs (DMCs) showed 14% of homology to known transcripts of *Paspalum notatum* reproductive transcriptome, and almost half of them were also differentially expressed between apomictic and sexual samples. DMCs showed homologies to genes involved in flower growth, development, and apomixis. Moreover, a high proportion of DMCs aligned on genomic regions associated with apomixis in *Setaria italica*. Several stage-specific differential methylated sequences were identified as associated with apospory expressivity, which could guide future functional gene characterization in relation to apomixis success at diploid and tetraploid levels.

## 1. Introduction

Apomixis, asexual reproduction by seeds, has been thoroughly studied to elucidate the developmental pathways that allow producing maternal clonal seeds avoiding meiosis and fertilization [1]. This trait is considered a valuable tool as its introduction into agricultural crops would allow perpetuating superior hybrid combinations, accelerating breeding processes, reducing costs and time spent on the new cultivars generations [2]. Although apomixis is widely distributed in the angiosperms, it is not a common trait between cultivated species [3]. The understanding of the molecular bases underpinning apomixis will open the way for extending its agricultural applications.

Gametophytic apomixis is a genetically determined trait strongly associated with polyploidy and mainly found in the tetraploid cytotype [4,5]. Although several works proposed that apomixis is inherited as a single dominant locus [1,6,7,8,9,10,11], many obstacles have hindered identifying key apomictic genetic determinants [12]. In this context, multiple studies in *Paspalum* spp. reported that apomixis is controlled by a large genomic region (called Apomixis Controlling Locus, ACL) with restriction in recombination, including highly repetitive sequences and high frequency of meiotic abnormalities [13,14,15,16,17]. Moreover, sequence analyses of the ACL in *P. simplex* and *P. notatum* revealed the presence of coding and non-coding sequences, pseudogenes, repetitive elements and heavy cytosine methylation [17,18,19,20,21]. Notwithstanding, comparative expression analysis led to identifying genes controlling the different apomixis components [22,23,24]. For instance, genes homologous to *SOMATIC EMBRYOGENESIS RECEPTOR KINASE1* (*SERK1*) [25] were detected both in *Poa pratensis* and *P. notatum* as related to the initial steps of apomixis development [26,27]; the *QUI GON JINN* (a *MAP3K* gene member of the *YODA* family) was reported as necessary to form aposporous embryo sacs in *P. notatum* [28,29], and the *APOLLO* gene was associated with apomeiosis in *Boechera* [30]. Recently, genetic and functional analyses of the *PN_TGS1*-like gene (a plant-specific RNA S-adenosyl methyltransferase), in transgenic lines of *P. notatum* carrying an antisense construction, proved the influences of the RNA processing in the ovule, the gametophyte and possibly the embryo development, as well as with aposporous development [31,32]. In relation to the other apomixis developmental components, *ASGR*-*BABY BOOM*-*like* was related to parthenogenesis control in *Pennisetum squamulatum* [33,34], and the *ORIGIN RECOGNITION COMPLEX* (*ORC3*) gene, initially found in the apomixis genomic region of *P. simplex*, was associated with the control of the pseudogamous endosperm development [20].

As most apomictic polyploids are facultative and apomixis and sexuality coexist in the same plant [1,12,35], it has been proposed that apomixis emerged from the rearrangement of sexual developmental programs [5,36,37]. Accordingly, functional analysis revealed that sexual mutants produce phenotypes reminiscent of apomixis development as unreduced gametes and multiple embryo sacs formation [38,39,40,41]. Alternatively, a recent discussion proposed that apomixis might be anciently polyphenic with sex. Then, the switch between both reproductive modes might be controlled by an upstream epigenetic process [42]. In line with epigenetic involvement, mutants of *Arabidopsis* in *ARGONAUTE 9*, a gene related to the silencing of transposable elements (TEs) in female gametes, and of other genes related to small interfering RNA (siRNA) biogenesis, lead to the differentiation of multiple gametic cells [43,44]. In addition, mutants of the members of the RNA-directed DNA methylation (RdDM) pathway (*DMT102*/*DMT105*/*CMT3*, *DMT103*/*DRM1* and *CHR106*/*DDM1*) display similar phenotypes [38], supporting the notion that epigenetic regulators lead to the induction of apomixis elements in sexual plants [24]. Furthermore, experimental evidence from natural apomictic systems showed that chemical induction of DNA demethylation in aposporous *P. simplex* reduces the rate of parthenogenesis significantly [19]. Moreover, global methylation analysis in facultative diplosporous *Eragrostis curvula* revealed that variations in apomixis/sexual expression, produced either overtime or by stress conditions, came along with epigenetic modifications [45,46]. Exploration of Cytosine methylation patterns in di- and tetraploid cytotypes and in different reproduction modes of *Ranunculus kuepferi* suggest that, in addition to the confirmed correlation of methylation profiles with the ploidy level, a causal relationship exists between methylation patterns and the mode of reproduction [47,48]. A recent comparative analysis between sexual and apomictic floral sRNA components of *P. notatum* detected significant differences associated with the reproductive mode, including auxin metabolism, transport and signaling [49].

Another important aspect to consider is the strong relationship between apomixis expression and polyploidy, which has been largely discussed [42,50]. Carman proposed that allopolyploidy, which emerged after hybridization, induces changes or asynchrony in the expression of a gene involved in the sexual pathway, resulting in apomictic development [51]. Consequently, as some model systems, such as *Paspalum*, *Ranunculus* and *Boechera*, express apomixis at the diploid level [52,53,54,55,56], it has been proposed that most angiosperms may have an inherent potential to switch to apomixis [57]. In a recent review, Hojsgaard pointed out that polyploids could better regulate stress conditions than diploids. As environmental stress is supposed to be the main natural triggers of sexuality, higher ploidy levels would release plants’ potential for apomeiosis [57]. Notwithstanding, the effects of polyploidy on apomixis success are not yet clear. Polyploidization of diploid *P. rufum* and *P. notatum* genotypes trigger an increment in apospory expressivity and in apomictic seed formation [52,58,59]. Moreover, polyploidization per se would also induce genomic rearrangements and epigenetic modifications, which might modulate transcription and influence apomixis expression [17,45,60,61,62]. In *Eragrostis curvula* and *P. notatum*, MSAP (Methylation Sensitive Amplified Polymorphisms) comparative analysis between diploid and tetraploid related genotypes revealed that polyploidization induces the emergence of new epialleles, increase overall methylation level and result in modulation of gene expression [61,62].

Despite the strong link between apomixis and polyploidy, it was found that some species of the *Paspalum* genus were able to produce aposporous embryo sacs (ES) at the diploid level [59,63,64,65]. *Paspalum rufum* Nees is a robust, erect perennial grass native to Paraguay, southern Brazil, Uruguay and northeastern Argentina [58], which is organized in an agamic complex composed of sexual diploids (2n = 2x = 20) and facultative aposporous tetraploids (2n = 4x = 40), [66]. However, previous works allow us to identify some natural diploid genotypes of *P. rufum*, producing viable ES and even complete apomixis under specific pollination conditions [52,55]. Then, sexual and apomictic development could occur in the same ovule. Apospory initials (AIs) emerge from nucellar cells along with tetrads generated after MMC meiosis, which, through megaspore and megagametophyte development, renders in many cases, meiotic and aposporous embryo sacs [67]. Moreover, hybridization between diploid genotypes generated both highly aposporic and highly sexual full siblings [68].

Here, we report an exploratory study on DNA methylation patterns associated with apomixis development in diploid *P. rufum* genotypes by methylation content-sensitive enzyme double-digest restriction-site-associated DNA (MCSeEd) [69,70]. This strategy was the best for our *P. rufum* system as it is a genome-independent approach with reduced sequencing demands and quite affordable in price [69,70,71]. Additionally, it allows analyzing different methylation contexts, CG, CHG and CHH, which is an important aspect to consider when focusing on plant epigenetic modifications. In particular, the relative methylation changes were tested on genomic DNA collected from full spikelets at two different developmental stages. Considering that most of the comparative studies of genomic methylation patterns between apomixis and sexual samples also included ploidy variations, we propose to compare two groups of diploid full sibling samples with contrasting apospory expressivity, avoiding both ploidy changes and high genetic variability. Methylation changes detected between samples clearly discriminate them according to their reproductive behavior. Functional analysis of differential methylated contigs (DMCs) showed their relation to genes and pathways involved in flower growth and development and even to some of those genes previously associated with apomixis development. Moreover, differential methylation sequences and context were quite specific to each of the developmental stages analyzed. As no reference genome is available, in silico mapping on to a very closely related species, *Setaria italica*, has also revealed that a higher proportion of DMCs is quite guided to genomic positions that were previously associated with apomixis.

## 2. Results

### 2.1. Sequencing Results

In this study, MCSeEd was applied to infer differences in DNA methylation associated with apospory expressivity in a diploid system of full siblings’ samples of *P. rufum* with contrasting modes of reproduction. Thirty-six MCSeEd libraries corresponding to three samples of each reproductive mode (highly aposporic and highly sexual), at two developmental stages (premeiosis/meiosis and postmeiosis), and for three methylation contexts (CG, CHG, CHH), were sequenced with Illumina HiSeq 1500 technology. This procedure generated a total of 151.74 million reads, of which 88% passed the quality control (QC > 30), allowing to obtain a total of 133.90 million reads of an average length of 143 pb (Appendix A). After demultiplexing, a total of 37.37 and 36.80 million reads were obtained for the highly sexual and highly aposporic groups, respectively, at premeiosis/meiosis. Likewise, 25.17 and 34.54 million reads were obtained for highly sexual and highly aposporic groups, respectively, at the postmeiosis stage (Appendix A).

### 2.2. Identification of Differentially Methylated Sequences

As no *P. rufum* reference genome is yet available, we used the MCSeEd bioinformatics pipeline to generate a pseudo-reference genome to map the filtered reads as previously described by Marconi et al. (2019) [69]. Reads from all samples at the same developmental stage (both highly sexual and highly aposporic) were processed together, and then two assemblies were obtained, one for premeiosis with a total of 1.18 million of contigs and another for postmeiosis with 1.09 million contigs (Table 1).

The proportion of contigs, sequenced and aligned, for each context were similar at both developmental stages. The CG context showed the highest number of contigs, and it accounted for more than 50% of the total for each developmental stage. CHH context presented an intermediate value of around 28%, while CHG context showed the lowest number of contigs with 15.18% at the premeiosis and 17.34% at the postmeiosis stage. After normalization, contigs with more than 10 reads were retained to estimate DMCs between highly aposporic samples compared to highly sexual samples in each developmental stage. Highly sexual samples were grouped and were compared with the bulk of highly aposporic ones. At premeiosis, from a total of 17,822 DMCs (Table 1 and Appendix A), 16,638 (93.36%) corresponded to asymmetric CHH context, while 826 (4.63%) and only 358 (2.02%) to CHG and CG contexts, respectively. At postmeiosis, a total of 31,618 contigs was found to be differentially methylated, almost doubling those found at premeiosis (Table 1 and Appendix A), of which 17,676 (55.90%) corresponded to CHG context, 11,855 (37.49%) to CG and 2087 (6.60%) to asymmetric CHH context. Then, in the early developmental stages, from MMC differentiation up to meiosis fulfillment, CHH context was the context more affected by differentially methylation between highly sexual and highly aposporic samples, while after meiosis and during megagametophyte development, both CHG and CG context predominated.

Principal component analysis (PCA) of DMCs at each methylation context was performed to evaluate if grouped samples clustered following DMCs methylation levels. At premeiosis, the first principal component (PC1) accounted for 60.7%, 67.9% and 62.1% of the total variation for CG, CHG and CHH contexts, respectively (Figure 1A–C). At postmeiosis, similar results were obtained, and PC1 explained 51.1%, 62.4% and 66.7% of the total variation in the three contexts CG, CHG and CHH, respectively (Figure 1D–F).

PCA showed clear discrimination between reproductive behaviors following methylation patterns. A complete linkage clustering based on the relative methylation levels at the DMCs was performed for each methylation context and developmental stage (Figure 2). At premeiosis, analysis of CHG and CHH context showed clear discrimination between reproductive modes, while for CG context, relative methylation of DMCs was not enough to clearly separate samples (Figure 2A–C). At postmeiosis, according to PCA analysis, methylation frequency at all methylation contexts allowed to cluster samples with similar reproductive behavior (Figure 2D–F). Moreover, DMCs were hierarchically clustered in two major groups showing the different methylation patterns (hypermethylated: yellow, or hypomethylated: red) (Figure 2D–F). Considering global methylation variations as related to the change from sexual to aposporous reproduction, we observed that the effect of apospory differed between developmental stages.

In premeiosis, the proportion of hypermethylated DMCs in aposporous samples was lower than hypomethylated, 0.6-fold and 0.7-fold, for CG and CHH context, respectively, but was higher 2.1-fold, for CHG context (Table 1, Appendix A). At postmeiosis, higher hypermethylation (1.4-fold) was observed in aposporous samples for CG context, but similar proportions between hypermethylated and hypomethylated DMCs were detected for CHG and CHH contexts (Table 1, Appendix A). All above suggest that global methylation differences related to developmental processes occurring along sporogenesis and gametogenesis are associated with differential apospory expressivity.

### 2.3. Identification of DMCs in Paspalum Notatum Floral Transcriptome

Given that the floral transcriptome of a very closely related species, *Paspalum notatum*, was just published [72] and to identify if DMCs were related to transcripts expressed during reproductive tissue development, a BLASTn search was performed on the available transcriptome. About 14.72% (7278) of DMCs showed to be similar to sequences belonging to the *P. notatum* floral transcriptome (Global Transcript Assembly, GTA_pn) database. DMCs at CHG context showed higher percentages of homology to GTA_pn transcripts than those belonging to CG and CHH contexts (18%, 10% and around 14%, respectively). These percentages were similar for both developmental stages (Table 2 and Appendix A and Section 3).

Moreover, as a quantitative comparison between transcripts (GTA_pn) from apomictic and sexual floral samples of *P. notatum* was also available [72], it was possible to find out which DMCs were similar to differentially expressed GTA_pn between apomictic and sexual samples (log_2_FC >|2| and padj <0.05). A total of 3127 (40%) of DMCs, out of the total homologous to GTA_pn, also corresponded to differentially expressed transcripts (Table 3). For all stages and contexts, similar proportions of DMCs associated with differentially expressed GTA_pn transcripts were detected (between 40 and 48%). Only CG context at premeiosis showed a lower percentage (28%).

### 2.4. Functional Analysis of Differentially Methylated Sequences

To infer which genes contain differential methylated sequences between samples, DMCs were also used as queries in BLASTn searches against the NCBI NT database.

To increases the number of identities, NCBI annotations of GTA_pn sequences detected in the previous section were also included. From the total amount of DMCs, 13.19% showed some similarities to sequences belonging to species related to *P. rufum,* such as *Paspalum plicatulum*, *Paspalum vaginatum*, *Paspalum minus*, *Paspalum ionanthum*, *Panicum hallii*, *Zea mays*, *Tripsacum dactyloides*, *Sorghum bicolor*, *Setaria italica*, *Setaria viridis* (Figure 3 and Appendix A, Section 1). Both at premeiosis and postmeiosis, CHG-DMCs showed a higher percentage of homology (18%) than the other two contexts, which showed between 9 and 11% of homology for both developmental stages (Table 2 and Appendix A and Section 1 and Section 4). Altogether, more than 77% of DMCs, that showed homology to NCBI NT database sequences corresponded to transcripts identified also in the *P. notatum* floral transcriptome (Appendix A). The remaining DMCs were homologous to molecular markers of very closely related species, such as microsatellite sequences of *Paspalum plicatulum* and *Paspalum vaginatum* (Appendix A). For instance, one of the CHH-DMCs identified at premeiosis significantly matched with the PnMA243 RAPD marker, which is genetically linked to apomixis in *P. notatum* [13] (Appendix A, Section 1).

Focusing on the genetic NCBI identity of mRNAs and CDSs, most of the functional annotations were involved in plant growth and reproductive development. Among which are: (i) **meiosis:** protein MEI2-like 4, protein DMC1 homolog B, MEIOTIC F-BOX protein MOF-like, CYCLIN-L1-1; (ii) **mitosis**: CYCLIN-G-associated kinase, CYCLIN-DEPENDENT KINASES G-2 A, B2-1, C-3, CYCLIN-A2-1-like, CYCLIN-T1-3, CYCLIN-T1-4, CYCLIN-B1-1-like, CYCLIN-D3-1, CYCLIN-P4-1-like, transcription factor MYB124, transcription initiation factor TFIID subunit 1, CELL DIVISION CYCLE 5-like, CELL DIVISION CYCLE 20.2, cofactor; (iii) **ubiquitination process**: E3 ubiquitin–protein ligases as WAV3-like (root gravitropism also expressed in leaf primordial), RHF2A and BIG BROTHER; (iv) **hormonal signaling**: auxin response factors (ARF3, 4, 11, 12, 16, 17, 18, 19, 22, 23), auxin-responsive proteins (SAUR36-like, IAA6, IAA21, IAA18) and auxin transport proteins (BIG); (v) **transcriptional regulation related with reproductive development**: transcriptional corepressor LEUNIG, SEUSS-like, trihelix transcription factor ASR3, ASIL2-like, MADS-box transcription factor 50 and 21, WRKY transcription factor 2, transcription factor KAN2, homeobox protein KNOTTED-1 homeobox protein *LUMINIDEPENDENS*, WUSCHEL-related homeobox; (vi) **epigenetic related processes**: S-ADENOSYL AND HISTONE METHYLTRANSFERASES, DNA-directed RNA POLYMERASE V subunit 1, INCREASED DNA METHYLATION 1-like and protein RNA-directed DNA methylation 3-like (Appendix A, Section 1).

As DMC corresponding to the CHH context were mainly present at premeiosis and associated with Doppia transposase genes of maize (Appendix A, Section 1), we mined for transposon-related sequences over the PGSB repeat element database. A total of 109 DMCs showed homology to 98 different transposon-related sequences mainly belonging to *Zea, Sorghum* and *Oryza* genera (Table 4 and Appendix A). As shown in Table 4, at premeiosis, most of the DMCs at CHH context (63) that were homologous to TEs were hypermethylated (40). At postmeiosis, 21 out of 44 DMCs associated with TEs, belonged to CHG context and the other 15 and 8 to CG and CHH contexts, respectively.

### 2.5. Detection of Differentially Methylated Genes Related to Apomixis

To further examine DMCs potentially related to apomixis, we run BLASTs searches over a specific database, compiled by Podio et al. (2020) [72], with a set of genes previously reported as genetically linked, differentially expressed or functionally associated with apomixis. Interestingly, 64 DMCs, 40 in premeiosis and 24 in postmeiosis, showed high similarity to apomixis-related genes (Appendix A). In particular, these DMCs matched with eight and six genes or hypothetical proteins in premeiosis and postmeiosis, respectively. Out of the five genes specific to premeiosis, we found: (i) N13 lncRNA of *P. notatum* (KX900513.1) [73] that was hit by two DMCs (PmCHH_E355570_L143 and PmCHH_E355576_L143) at CHH context; (ii) three *APOLLO* alleles of *Boechera* spp.[30], two apomictic (KF705603.1, KF705605.1) and one sexual (KF705604.1), that showed homology to four DMCs (PmCHH_E465722_L143, PmCHH_E580026_L143, PmCHH_E849538_L143, PmCHH_E760051_L143) at CHH context.

Other three genes were found to be specific to postmeiosis (Appendix A) (i) the *Poa pratensis ANKYRIN PROTEIN KINASE-LIKE* (AJ810709.1) [74,75] that matched with two DMCs at CHG (PoCHG_E105508_L143, PoCHG_E144640_L143); (ii) *ORC3c* pseudogene of *P. simplex* (LN832400.1) [20] that hit by a CHG-DMC and (iii) *APOSTART1* gene of *P. pratensis* (AJ786392.1) [26,76] that matched with a CHG-DMC (PoCHG_E110825_L143) (Appendix A).

Finally, two genes and one hypothetical protein were common to both stages: (i) *SERK2* of *P. pratensis* (AJ841697.1) [26,27] that was tagged by one CHG-DMCs at premeiosis (PmCHG_E216648_L143) and two at postmeiosis (PoCHG_E176839_L143, PoCHG_E100557_L143, and (ii) *ATP-DEPENDENT DNA HELICASE* of *P. simplex* (MH106546.1) [77], that matched at several locations with 12 DMCs, 11 CHH-DMCs and 1 CHG-DMC, at premeiosis and two CHH-DMCs at postmeiosis. Moreover, when a lower identity value was considered, another seven different fragments of *SERK2* matched with eight additional DMCs, mainly from postmeiosis, belonging to the three contexts and 26 different fragments of *ATP-DEPENDENT DNA HELICASE* aligned to 70 additional CHH-DMCs, mainly at premeiosis (61 DMCs) (Appendix A). Therefore, we can summarize that some of the genes previously identified as associated with apomixis are highly differentially methylated between highly aposporic and highly sexual genotypes of *P. rufum*.

### 2.6. GO and KEGG Analyses of Differentially Methylated Contigs

A Gene Ontology (GO) classification for all DMCs was performed using the Arabidopsis database; homologies from *P. notatum* transcriptome were also included (Appendix A, Section 2). From overall DMCs, 8.53% showed a significant homology; at premeiosis, 14%, 5% and 6% of CHG-, CHH- and CG-DMCs, respectively, were homologous to sequences at the *Arabidopsis* database. While at postmeiosis, 12%, 7%, and 6% of CHG-, CG-, and CHH-DMCs, respectively, were found. Therefore, in both stages, CHG-DMCs showed a higher percentage of homology with *Arabidopsis* genes than DMCs of the other contexts (Table 2 and Appendix A
Section 2). Moreover, 4205 (8.5%) DMCs were grouped into 91 GO terms in the following categories: 24 cellular components (CC), 25 biological processes (BP) and 42 molecular functions (MF). A GO enrichment analysis was also performed for the DMCs at both developmental stages (i.e., premeiosis and postmeiosis). 

Figure 4A–C shows the 20 most representative GO terms for each category (*p* < 0.03), specific GO terms were found to be differentially methylated in each analyzed stage. The most represented GO terms in the CC category, both at premeiosis and postmeiosis, were ubiquitin ligase complex, endosome and cytoskeleton, while nuclear lumen and Cul4−RING E3 ubiquitin ligase complex was specific to premeiosis. Vacuolar membrane, vacuolar part Golgi apparatus, Golgi subcompartment and transferase complex were found enriched only at the postmeiosis stage (Figure 4A). Regarding the BP category, most of GO terms were common to both stages and were related to meristem and flower growth and development, such as regulation of the reproductive process, regulation of flower development, vegetative to the reproductive phase transition of the meristem, metabolic process, regulation of shoot system development, meristem structural organization, meristem maintenance, pattern specification process, regulation of post−embryonic development and mitotic cell cycle. Additionally, mRNA metabolic process, gene silencing, gene silencing by miRNA and production of miRNAs involved in gene silencing by miRNA were significantly represented at both stages. At premeiosis, some terms showed a higher enrichment than those at postmeiosis, like protein modification by small protein conjugation and protein ubiquitination, and those BP related to actin filament organization tropism, gravitropism and response to gravity were only present at postmeioisis (Figure 4B). According to the MF category, the principal enriched common terms included protein serine/threonine kinase activity, adenyl nucleotide and ribonucleotide-binding and ATP-binding. Regarding developmental stages, premeiosis showed low numbers of highly represented enriched terms as ubiquitin−like and ubiquitin-protein transferase activity. Double-stranded RNA-binding was also specific but not highly represented. At postmeiosis, ATPase activity, phosphoric ester hydrolase activity and active transmembrane transporter activity were highly represented specifically at this stage (Figure 4C).

On the other hand, according to the KEGG analysis, the pathways involved in the reproductive process were also inquired. KEGG pathway annotation was possible for 1724 DMCs that belong to 137 pathways (Appendix A, Section 2). The KEGG enrichment analysis showed specific terms for both premeiosis and postmeiosis stages (Figure 4D). In the youngest stage, endocytosis, spliceosome, ubiquitin-mediated proteolysis and glycolysis/gluconeogenesis are the most represented, while at postmeiosis, RNA degradation, phagosome and glycerolipid metabolism were present. Only glycerophospholipid metabolism was common in both stages.

As previous works revealed, AIs, in the diploid *P. rufum* cytotype, are formed mainly during megasporogenesis, from MMC meiosis up to functional megaspore differentiation [67]. To further understand the pathways that could be affected by epigenetic regulations during AI development, we decided to select those genes that were specifically differentially methylated during premeiosis and analyzed their interaction using the STRING platform. Then, *Arabidopsis* genes orthologous to DMCs were initially analyzed by Venn diagrams to look for putative genes specifically modified in each of the developmental stages.

From a total of 749 genes at premeiosis and 2504 at postmeiosis, only 321 were common to both stages, while 482 and 2183 were specifically differentially methylated at premeiosis and postmeiosis, respectively (Figure 5A). The associated network among premeiosis-specific genes (Figure 5B) showed that most of them were connected. The STRING network notably showed a central node that guides almost all the connections. This central gene, *CELL DIVISION CYCLE 5-LIKE* (*CDC5*, AT1G09770), was associated with members of processes like replication, transcription, translation and post-translation processes. Interestingly, DNA replication (Figure 5B, pink-tagged), mitosis as well as chromosome-associated proteins involved in nucleosome assembly were detected (Figure 5B, orange-tagged). Two genes previously reported as components of apomixis cascade were connected, *FIE* (AT3G20740), [78] and *ORC2* (AT2G37560) [20]. Additionally, a high number of genes involved in RNA metabolism like RNA synthesis, degradation, transport and posttranscriptional silencing (Figure 5B, lilac) as well as those related to splicing were also connected (Figure 5B, yellow-tagged). Additionally, a set of genes involved in protein translation and post-translational regulation as a ribosome pathway, translation factors (Figure 5B, light blue and red-tagged, respectively), ubiquitin system and membrane trafficking (Figure 5B, green-tagged) were also found.

### 2.7. Mapping DMCs on Setaria Italica Genome

As no reference genome is still available for *P. rufum* or other *Paspalum* species, we mapped DMCs on the *Setaria italica* genome, a closely related species to *P. rufum* [79], with the aim of mapping the DMCs. In silico mapping showed that out of the total DMCs, 2154 (12.1%) for premeiosis and 5836 (18.4%) for postmeiosis aligned to the *S. italica* genome (Appendix A, Figure 6).

When we analyzed the distribution of DMCs in each *S. italica* chromosome, at premeiosis, the average number of DMC per chromosome was 239.3 (with a minimum of 116 and a maximum of 396 at chromosomes 8 and 5, respectively), while at postmeiosis, the average number of DMCs per chromosome was 648.4 (with a minimum of 235 and a maximum of 996 at chromosomes 8 and 5, respectively) (Appendix A, Figure 6). It is worth noting that although DMCs are distributed all along the genome, chromosomes one, three and five had a higher relative number of matches compared to the rest of the chromosomes.

## 3. Discussion

Here we implemented an innovative reduced-representation and reference-free MCSeEd approach for characterizing genome methylation patterns across different methylation contexts [69,70] by comparing highly aposporic and highly sexual diploid hybrids samples of *P. rufum* [68]. This study was aimed at finding whether differential methylation patterns were related to apospory development at the diploid level. Our strategy has practical advantages as it reduces sequencing demands, avoids reference genome availability and allows the analysis of three methylation contexts (CG, CHG and CHH). Moreover, it also presents reduced genetic polymorphism due to the comparison of full-sib samples with no ploidy differences that often occur in the analysis of apomictic species.

Therefore, two groups of diploid samples, highly aposporic and highly sexual, at two developmental stages were analyzed and compared. The total number of contigs obtained from the pseudo-reference genome was similar for both developmental stages (around 1 million each). Particularly, the CG context accounted for more than 50%, followed by CHH context with 28% and CHG with around 15% at both developmental stages. At the same time, total DMC were about 1.5% and 2.9% at each developmental stage, premeiosis and postmeiosis, respectively. These values, such as total contigs, contigs per context and DMC, were comparable to those obtained by the reference-free MCSeEd approach in maize and globe artichoke [69,71]. When total DMCs were clustered, most of them grouped samples by reproductive modes, indicating that the discrimination between highly aposporic and highly sexual samples mainly resided in the differential methylation patterns. Within each reproductive mode, some variations were anyway recorded, which may be due to genetic differences between genotypes. Only CG context at premeiosis failed to cluster samples by reproductive behavior. Regarding the latter, methylation analyses based on restriction enzymes have been extensively performed in apomictic non-model plants; all were focused on CG context. Generally, these results revealed that apomixis expression is related to epigenetic variations. However, differential methylation patterns are not exclusively related to reproductive behavior, and polyploidization per se also induces epigenetic modifications [45,61,62,80]. In *P. notatum*, a preliminary MSAP analysis at CG context clustered together apomictic tetraploid genotypes and one sexual tetraploid accession was affiliated with sexual diploids, while another experimentally obtained sexual tetraploid was out of both groups [62]. Even when only tetraploid genotypes of *P. notatum* were included, clustering analysis by epigenetic similarities revealed that sexual genotypes were grouped closer than apomictic ones clustered both with apomictic and sexual samples [80]. Similar studies in *Eragrostis curvula* reported that the CG methylation status clustered samples by ploidy level but not by reproductive behavior [61]. Then, the fact that all, but CG-DMC at premeiosis, allowed clear discrimination between reproductive modes could be assigned to the increment on the number of loci analyzed (one or two orders of magnitude higher) compared to previous MSAP analysis, the homogenous genetic background and ploidy level between samples and the restricted tissues and developmental stage selected.

Focusing on the global methylation changes associated with highly aposporic samples, such as hypo- or hypermethylation, we observed not a unique way of variation for all the contexts and samples analyzed, but a different behavior for each one. Contrasting and opposite effects were recorded for CG and CHG contexts. While at premeiosis apospory appeared to reduce methylation in the CG context and increase it in the CHG context, the opposite sense occurred at postmeiosis. Only at CHH context apospory seemed to essentially reduce methylation levels at both stages analyzed. Contrary to our results, previous works in *E. curvula* and *P. notatum* showed similar general methylation status between apomictic and sexual genotypes. However, these investigations were restricted to CG methylation, did not consider different flower developmental stages and were based on a low genome coverage technique [61,80].

It is worth noting that at premeiosis, the CHH context accounted for more than 90% of the DMCs compared to CG and CHG contexts. Regarding development, at this stage, female sporogenesis occurs, and MMC undergoes meiosis, while in aposporic genotypes, AI would also emerge from nucellar cells [67]. Then, our observations agree with previous studies reporting that in contrast to symmetric methylation contexts, CHH methylation is highly labile and reprogrammed by demethylation/remethylation during female sporogenesis, showing reduced methylations at MMC, which is gradually recovered up to meiosis onset, while CG methylation is globally maintained in female gametic precursors [81]. Accordingly, during these developmental stages, chromatin is gradually decondensed by reducing heterochromatin, which is also characterized by high CHH methylation levels [82]. Therefore, differences in developmental processes occurring between highly aposporic and highly sexual samples, related to megasporogenesis and AI differentiation, could be reflected by differential methylation at CHH context and chromatin remodeling. Moreover, our BLAST searches revealed that factors influencing chromatin state as transposon-related sequences, histone methyl- and acetyltransferases, histone 3.3 and ATP dependent DNA helicases (SUVH1 and HAC) [83,84], were related to DMCs preferentially hypermethylated at CHH context at this stage of development. Genetic evidence indicates that RNA-dependent DNA methylation (RdDM), which produces de novo CHH methylation at transposon edges, euchromatin, or at heterochromatin/euchromatin boundaries, might also influence megasporogenesis [85,86] as it was found to be involved in MMC specification and differentiation [43,87]. Moreover, recent comparative transcriptome analysis of *P. notatum* and *Hypericum perforatum* L. aposporic model systems found several genes involved in RdDM as differentially expressed between apomictic and sexual genotypes [49,88]. Our results leave the challenge to find out the interplay between the RdDM pathway, CHH methylation and sexual to apomixis switch. Otherwise, during the postmeiosis stage, *P. rufum* ovules undergo megagametogenesis, and in highly aposporic samples, unreduced AESs are also developed beside the sexual ES [67,68]. It has been recently reviewed that at this stage, epigenetic dynamics are not so clear. However, many studies have reported aspects related to it [89]. For instance, the analysis of DNA methylation mutants of *Arabidopsis* showed that different mechanisms are involved in establishing gamete-specific epigenetic patterns in the egg and central cells [90]. Another study proposed that the somatic sRNA pathway involving *ARGONAUTE5* promotes megagametogenesis [91] and genome-wide analysis of DNA methylation in egg and central cells of rice and *Arabidopsis*-reported locus-specific active demethylation in the central cell [92]. Therefore, it is not surprising that different developmental processes occurring during megagametogenesis in both types of samples may result in differential methylation profiles.

As there is a strong correlation between DNA methylation and silencing of transposons [86] and this type of repetitive element is present in the apomixis locus of many species [18,93,94,95], we mined DMCs for homologies to repetitive elements. Accordingly, we found that DMCs were associated with transposons, mostly retrotransposons, at both developmental stages and for all methylation contexts. Particularly, hypermethylated CHH-DMCs prevailed at premeiosis in the highly aposporic group, but hypomethylated DMCs were enriched at postmeiosis in all methylation contexts. Some earlier reports suggest that in *P. notatum,* some transcripts related to retrotransposons are differentially expressed between apomictic and sexual plants [96,97]. Moreover, a higher rate of CG methylation and a higher expression of retrotransposon transcripts was reported in sexual genotypes of *E. curvula* when compared to apomictic ones [45]. Otherwise, a recent work carried out in *Cenchrus ciliaris* showed that transcripts of Gy163, an LTR retrotransposon, are more expressed in the reproductive tissues of apomictic than of sexual plants, and this was correlated with reduced methylation level [98]. A recent study about the role of miRNA-mRNA interactions in apomixis development in *E. curvula* found that a transposon sequence was specifically repressed in the sexual genotype, most likely due to interactions with miRNAs [99]. Therefore, as differential expression of TE-related sequences could be assigned to differential methylation and epigenetic regulations, which would influence apospory expressivity, further specific analysis of DMCs related to LTR retrotransposons deserve to be performed in future works.

Since no reference genome was available, it was not possible to determine the position of DMCs in terms of gene-body-extended sequences. However, the floral transcriptome of *P. notatum* has just been released [72] and was used in this study. Although the two species belong to different taxonomic groups [100,101,102], phylogenetic analysis, including non-coding cpDNA fragments and morphological data, grouped them in the same clade [103]. Consequently, intergenic and regulatory genomic sequences beyond transcribed regions were missed in the present analysis. Notwithstanding, fourteen percent of total DMCs were aligned to *P. notatum* floral transcriptome. This proportion is comparable to similar analysis reported by previous application of the MCSeEd approach [69]. Notably, the percentage of CHG-DMC that matched with transcripts differentially expressed between apomictic and sexual genotypes of *P. notatum* was higher than DMCs belonging to other contexts. Accordingly, high enrichment of CHG methylation in exons was observed by Marconi and collaborators in maize under water stress conditions [69]. This evidence could be related to the fact that *CHROMOMETHYLASE 3* (*CMT3*), which is involved in CHG methylation at gene-body, is downregulated in the ovules of apomictic plants [38,104,105]. Moreover, loss of *DMT102* activity in maize ovules, a gene related to *Arabidopsis CMT3*, revealed phenotypes reminiscent of apomictic development, resulting in transcriptionally competent chromatin state in the archesporial tissue and in the egg cell that mimics the chromatin state found in apomicts [38].

To have an overall view of genes that seem to be differentially methylated during premeiosis, when AIs start differentiating [67], we predicted protein interactions using the STRING platform. This analysis revealed a compact network, connecting almost all the members by a central node, suggesting that differential methylation affects genes functionally related during developing reproductive tissues. Indeed, the central gene, *CELL DIVISION CYCLE 5-LIKE* (*CDC5*, AT1G09770), is involved in different pathways as miRNA processing [106,107], G2/M cell cycle phase transition, and it was reported to regulate SAM maintenance by controlling the expression of *SHOOT MERISTEMLESS* (*STM)* and *WUSCHEL (WUS)* [108]. The network also revealed a connection with genes previously linked to apomixis, as *FIE* [78] and *ORC2* that interact strongly with *ORC3* previously reported to be genetically linked to apomixis in *P. simplex* [20]. These observations were reinforced by BLASTn searches of DMCs against NCBI NT, where additional genes related to apomixis as *ORC3b* [20], *ARGONAUTE 4A* [49,105], and LRR receptor kinase *SOMATIC EMBRYOGENESIS RECEPTOR KINASE-like* (*SERK2*) [26,27] were also detected. Moreover, other apomictic related genes were also-tagged by DMCs at postmeiosis, such as *ANKYRIN PROTEIN KINASE-LIKE* [74,75], *ORC3c* pseudogene [20] and *APOSTART1* gene of *P. pratensis* [26,76]. Interestingly, a BLAST analysis run years later of the original publication showed that the sequence, originally named PpAPK [74,75], is actually similar to a putative methyltransferase found in numerous Gramineae and best hit (98% similarity and 95% identity) with the “probable” methyltransferase PMT2 of *Triticum dicoccoides* (accession number XP_037434607.1). Given its differential expression [26] and differential methylation in relation to apomixis vs. sexual development, it makes this gene one of the best candidates for further investigations.

All the above show that several specific apomixis-related genes were tagged by DMC at each stage of development. This specificity was also evidenced on GO/KEGG classification of those *Arabidopsis* genes homologous to DMCs. Venn analysis, including more than 3000 *Arabidopsis* genes-tagged by DMC, showed that only 321 were common to both stages. This stage-specific differential epigenetic regulation agrees with the KEGG pathways prediction of those differentially expressed transcripts in apomictic *P. notatum* genotypes, where different molecular routes are specifically modulated at each developmental stage [72]. Moreover, some similarities were found between molecular routes specifically modulated at each developmental stage in *P. notatum* and those found in our work, such as glycolysis/gluconeogenesis, ubiquitin-mediated proteolysis, fatty acid degradation, sphingolipid metabolism, endocytosis, peroxisome, tyrosine metabolism and glycerophospholipid metabolism at premeiosis/meiosis stage and RNA degradation, glycerolipid metabolism and ABC transporters at postmeiosis [72]. Reinforcing our previous observations, GO classification was identified among the biological process category, terms related to gene silencing and regulation of gene expression by epigenetic, which were also enriched in *P. notatum* differentially expressed transcript [72]. Accordingly, ovule gene expression analysis in sexual and aposporous genotypes of *Hypericum perforatum* evidenced a differential and heterochronic expression of transcripts involved in epigenetic regulation of gene expression [88].

Since no reference is available for *Paspalum* spp. to map DMCs on the genome, we decided to carry out an in silico mapping on *Setaria italica*, a related species [79]. DMCs belonging to both stages of development were distributed along the nine *S. italica* chromosomes. However, by taking into account chromosome size, the relative proportion of DMCs was quite guided to chromosomes one, three and five. As previous works revealed a conserved synteny for the apomixis locus of *Brachiara decumbens*, *Brachiaria humidicola* and *Paspalum simplex* in chromosomes 1, 3 and 5 of *S. italica* [77,109,110] and in agreement with the homology between DMCs and many apomixis related genes and molecular markers, our results suggest that DMCs between highly aposporic and highly sexual samples are quite guided to genomic positions that were previously associated with apomixis.

## 4. Materials and Methods

### 4.1. Plant Material

Four diploid genotypes (F_1_#9, F_1_#12, F_1_#15 and F_1_#39), belonging to an F_1_ population previously obtained and established at the experimental field of Agronomy collage (National University of Rosario), were selected according to their mode of reproduction [68]. Plants F_1_#9 and F_1_#12, showing 1.22% and 0% of ovules bearing AES, respectively, were selected for conforming to the highly sexual group. Individuals F_1_#15 and F_1_#39, showing 32.7% and 35.8% of ovules bearing one or more AES, respectively, were chosen to constitute the highly aposporic group (Appendix A). Spikelets from the different genotypes were collected during one flowering season (2018) and frozen. Ovule reproductive developmental stages were determined as described by Soliman et al. 2019 [67]. Briefly, as male and female developments are synchronic, the stages of female development were inferred from observation of microsporogenesis/microgametogenesis. Anthers were dissected from spikelets and squashed over a drop of 2% (*w/v*) aceto-carmine. Then, the reproductive stage was scored under light microscopy [67]. Spikelets carrying anthers showing microspore mother cells (MiMC), dyads, and tetrads were considered as being at the premeiosis/meiotic stage (called premeiosis in advance to simplify). At this point, ovaries contain megaspore mother cells (MMC) and/or meiotic products [67]. Spikelets containing anthers exhibiting unicellular microspores and/or bicellular pollen grains were considered as being at the postmeiosis stage. At this stage, ovaries contain functional megaspores, some of which undergo megagametogenesis through three consecutive mitotic divisions [67].

### 4.2. DNA Extraction

DNA extraction was carried out using the CTAB method according to Martínez et al. 2003 [13] with the following modifications. Frozen plant tissue (100 mg of full spikelets) was powdered in Eppendorf tubes by using plastic pestles and incubated with 1.2 mL of extraction buffer (100 mM Tris-HCl pH 7.5; 700 mM NaCl; 50 mM, EDTA pH 8.0; 2% *w/v* CTAB; 140 mM 2-mercaptoethanol) with gentle agitation at 65 °C for 60 min. An equal volume of chloroform was added to each sample, mixed for 10 min, and centrifuged at 5000× *g* for 15 min. The aqueous phase was collected, and a new extraction with chloroform was carried out. Then DNA from the aqueous phase was precipitated by adding an equal volume of chilled isopropanol. DNA samples were centrifuged for 15 min at 10,000× *g*. The pellets were washed in 70% ethanol, air-dried, and resuspended in 200 ul of TE (1 mM Tris-HCl, 0.1 mM EDTA) pH 8.0. Samples were incubated in 1% of RNAse solution for 90 min at 37 °C, and after a dilution with ultrapure water to a final volume of 500 µL, were passed through a new chloroform step. The aqueous phase was collected, and DNA was re-precipitated by adding 5% *v/v* of 5 M NaCl and two volumes of chilled absolute ethanol. After overnight incubation at −20 °C, DNA was centrifuged (20 min at 10,000× *g*), washed in 70% ethanol, dried at 37 °C in an oven for 1 h, and dissolved in 200 uL of ultrapure water. DNA samples were quantified by using a Qubit ® fluorometer (Life Technologies™).

### 4.3. Library Construction and Illumina HiSeq Sequencing

The library set-up protocol was performed according to Marconi et al. (2019) and Di Marsico et al. (2020) [69,70]. DNA samples from full spikelets of four different genotypes corresponding to highly aposporic group (F1#9, F1#12) and highly sexual group (F1#15, and F1#39), plus one technical replicate (a duplication of one genotype for each group), were generated, to reach a total of three samples per reproductive mode (see Appendix A for a detailed description). Briefly, three specific enzyme combinations were chosen, including each of the three methylation-sensitive enzymes, combined with methylation-insensitive *Mse*I, to infer CG (*Aci*I/*Mse*I), CHG (*Pst*I/*Mse*I) and CHH (*Eco*T22I/*Mse*I) methylation contexts, respectively (Appendix A). A total of 36 libraries were constructed, considering three DNA samples per two reproductive modes (highly aposporic and highly sexual), at two developmental stages (premeiosis and postmeiosis) and at three methylation contexts (CG, CHG, and CHH). (Appendix A). Then, for each next-generation sequencing (NGS) library, 150 ng of purified DNA were double digested with one of the three enzyme combinations. In the same reaction, a sample-specific barcoded adapter (P1, P2, P3, P25, P26, P25, P31, P32, P33, Appendix A) was ligated to the methylation-sensitive restriction end (for sequence details, see Marconi et al. 2019), while a common Y adapter was ligated to the sticky end left by *Mse*I, following the protocol described by Marconi et al. (2019) [69]. The libraries were then pooled, as reported in the experimental design (Appendix A), purified using magnetic beads (Agencourt AMPure XP; Beckman Coulter, MA, USA), size selected by gel electrophoresis, and purified using QIAquick gel extraction kit (Qiagen) for fragments ranging from 250 bp to 600 bp. Size-selected libraries were quantified using a fluorometer (Qubit; Life Technologies), and a normalized DNA amount (15 ng) was amplified with a primer that introduced an Illumina index (at the Y common adapter site) for demultiplexing (Appendix A). Following PCR with uniquely indexed primers, multiple samples were pooled. PCR-enrichment was performed as described by Marconi et al. (2019) [69]. The grouped libraries were pooled in an equimolar fashion, and were Illumina-sequenced by HiSeq 1500 Illumina, using 150-bp single-end chemistry.

### 4.4. Reference-Free MCSeEd Protocol

The raw reads were checked by quality analysis using the FastQC program (www.bioinformatics.babraham.aC.uk/projects/fastqc/ accessed on 1 April 2021) employing a pipeline developed by Novogene Company. Briefly, adapter sequences, duplicate sequences, reads containing N > 10% (where N represents the base cannot be determined), ambiguous and poor-quality reads (with a base count of Phred value <20), were removed using the TrimGalore program (https://www.bioinformatics.babraham.ac.uk/projects/trim_galore). Since there is no reference genome for *P. rufum*, the assembly was done using the reference-free pipeline developed for MCSeEd: https://bitbucket.org/capemaster/mcseed/src/master/ [69]. Raw reads were assembled using Rainbow 2.0.4 (https://sourceforge.net/projects/biorainbow; using following algorithms: cluster, div and merge) and CDHit (https://github.com/weizhongli/cdhit; using following parameters: -mask N, -M 20,000, -c 0.95), to generate a pseudo-reference genome that consisted of a multi-fasta file containing the contigs. This was used as a guide for the mapping algorithm. After mapping the reads to this pseudo-reference-genome using the bwa mem algorithm with the default configuration, a count matrix was generated for each sample using SAMtools, which counts the number of sequences mapped in each contig. This matrix was used to quantitatively identify differentially methylated contigs [69].

### 4.5. Statistical Analysis

Statistical analyses were performed in R version 3.3.2 (www.r-project.org accessed on 1 April 2021) using the “stats”, “factoextra”, and “gplots” packages. The “stats” package was used to estimate correlations and binomial and logistic regression, and “factorextra” was used to perform the principal component analysis. Complete linkage clustering was carried out using the “heatmap.2” function of the “gplots” package, in combination with the “hclust” and “dist” functions, and with “ward.D2” as the clustering method. The MethylKit R package [111] was used to estimate the methylation changes between highly aposporic and highly sexual samples.

### 4.6. Sequence Deposition

Raw sequencing reads obtained in this work were deposited in the NCBI Short Read Archive under the following SRA accession: PRJNA708155.

### 4.7. Analysis and Functional Annotation

DMCs were blasted (BLASTN) on the NCBI NT database (https://www.ncbi.nlm.nih.gov accessed on 1 April 2021) (threshold values of %ID > 70, query cover > 90, *e*-value < E−15) to assign homologies with known sequences. A similar analysis was performed using BLASTx on *Arabidopsis* database available at the TAIR webpage (https://www.arabidopsis.org/download/) (% ID > 30, query cover > 90, *e* value < 1E−5). As the homologies of DMCs with NCBI and TAIR database was low, a search for *P. notatum* transcriptome database (NCBI accession: GIUR00000000) was also performed (% ID > 70, query cover > 90, *e*-value < 1E−15), to increase the number of identities for the DMCs. Likewise, DMCs were queried against the sequences of known transposable elements available at PGSB repeat element database (ftp://ftpmips.helmholtz-muenchen.de/plants/REdat/mipsREdat_9.3p_ALL.fasta.gz accessed on 1 April 2021) (% ID > 70, query cover > 90, *e*-value < 1E−5

### 4.8. GO Analysis and Pathway Mapping of DMCs

The GO numbers were obtained by using clusterProfiler (http://www.geneontology.org accessed on 1 April 2021) [112] over the *Arabidopsis* database (*e*-value 1E−6). Moreover, DMCs were analyzed with the Kyoto Encyclopedia of Genes and Genomes (KEGG) pathways database (https://www.genome.jp/kegg/ko.html accessed on 1 April 2021) [113,114,115] and critical differentially expressed pathways were analyzed by STRING (https://string-db.org accessed on 1 April 2021) [116] to infer their possible function and association between them.

### 4.9. In Silico Mapping

The DMCs were aligned to *Setaria italica* genome (Setaria_italica_v2.0, GCA_000263155.2) with Bowtie2 v2.3.2 algorithms using the following settings: -N 1 -L 15 --very-sensitive-local -k 5. The proportions of DMC, relative to the total Mpb of each chromosome, were calculated.

## 5. Conclusions

Our work establishes the first reduced-representation epigenetic analysis of DNA methylation in reproductive tissue of diploid *P. rufum*. Overall, it exposed that apospory expressivity is related to differential methylations patterns in CG, CHG and CHH contexts. Moreover, our findings suggest that several genes, previously associated with reproductive development, reproduction and apomixis, appeared affected by developmental stage-specific epigenetics marks that could regulate the reproductive behavior and the level of apospory expressivity in *P. rufum*. This dataset will allow an efficient selection of apomixis candidate genes for further analysis to detect the switch that controls apomixis expressivity in a diploid environment.

## Figures and Tables

**Figure 1 plants-10-00793-f001:**
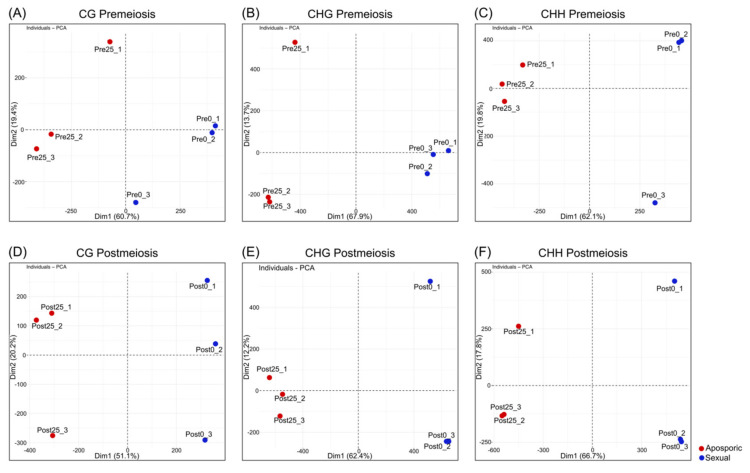
Principal component analysis based on the relative methylation levels at the DMCs obtained for each replicate and methylation context. (**A**−**C**) Premeiosis (Pre) stage and (**D**−**F**) postmeiosis (Post) stage. (**A**,**D**) CG context; (**B**,**E**) CHG context; (**C**,**F**) CHH context. Numbers in brackets indicate the fraction of overall variance explained by the respective component (Dim1, Dim2). Blue dot: highly sexual samples, red dot: highly aposporic samples.

**Figure 2 plants-10-00793-f002:**
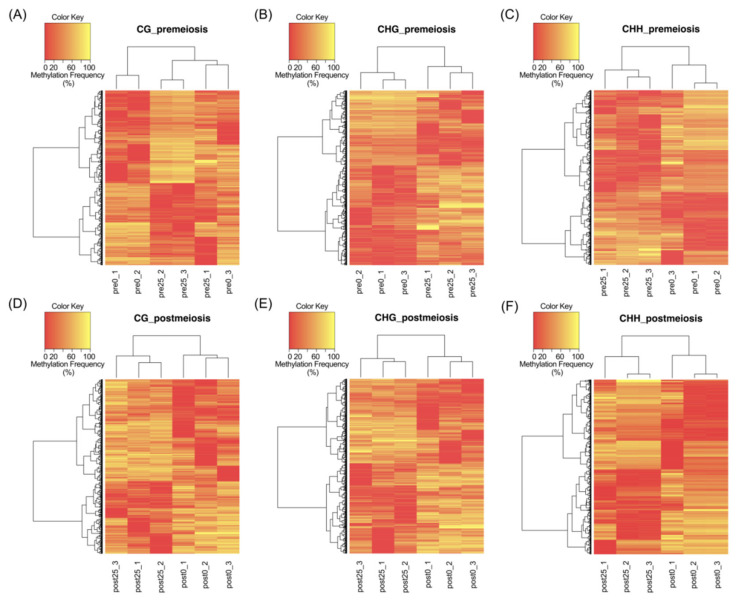
Complete linkage clustering based on the relative methylation levels at the DMCs, in each context of all samples. (**A**–**C**) Premeiosis (Pre) stage and (**D**–**F**) postmeiosis (Post) stage. (**A**,**D**) CG context; (**B**,**E**) CHG context; (**C**,**F**) CHH context of highly sexual (0) and highly aposporic (25) samples, respectively. Colors indicate the level of methylation for each position, being yellow as hypermethylated and red as hypomethylated.

**Figure 3 plants-10-00793-f003:**
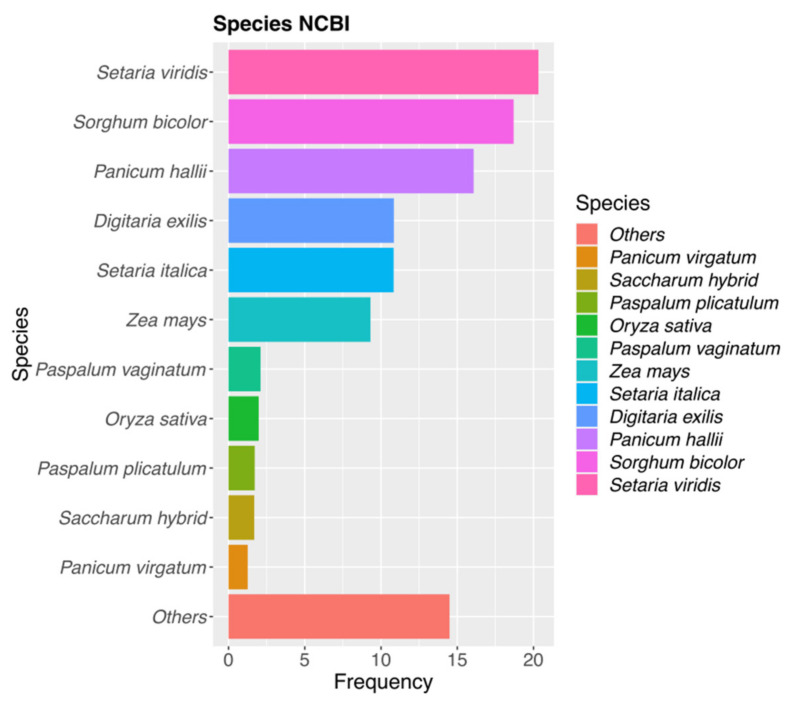
Species distribution of top hits revealed by the DCMs vs. NCBI-NT database. Most of the hits corresponded to Gramineae species and matched better with *Setaria viridis* and *Sorghum bicolor*.

**Figure 4 plants-10-00793-f004:**
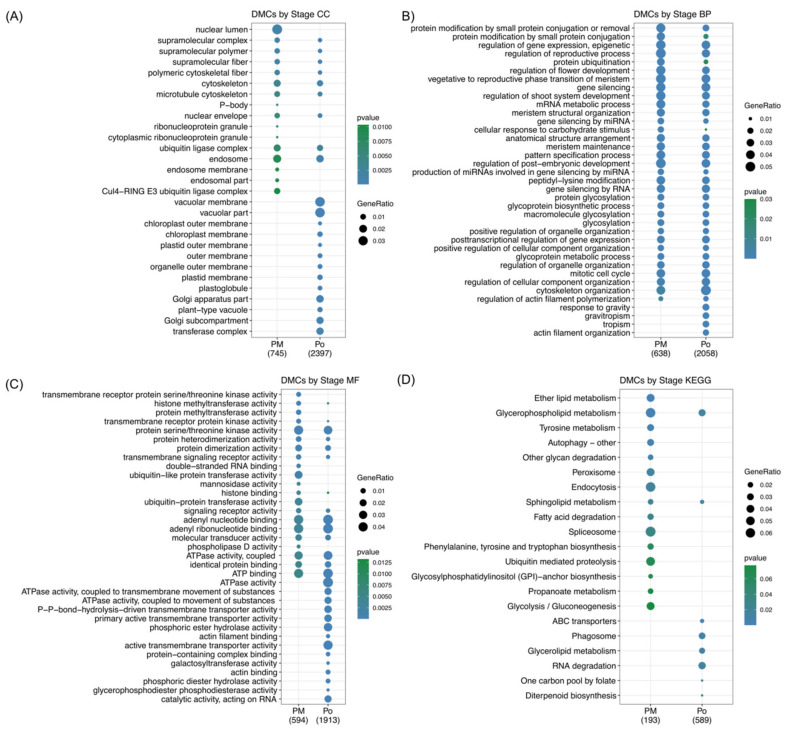
Gene Ontology (GO) and Kyoto Encyclopedia of Genes and Genomes (KEGG pathways enrichment analysis comparing premeiosis and postmeiosis stages. (**A**–**C**) 20 most representative GO terms for each category. CC: cellular component. BP: biological process. MF: molecular function. (**D**) 20 most represented KEGG pathways.

**Figure 5 plants-10-00793-f005:**
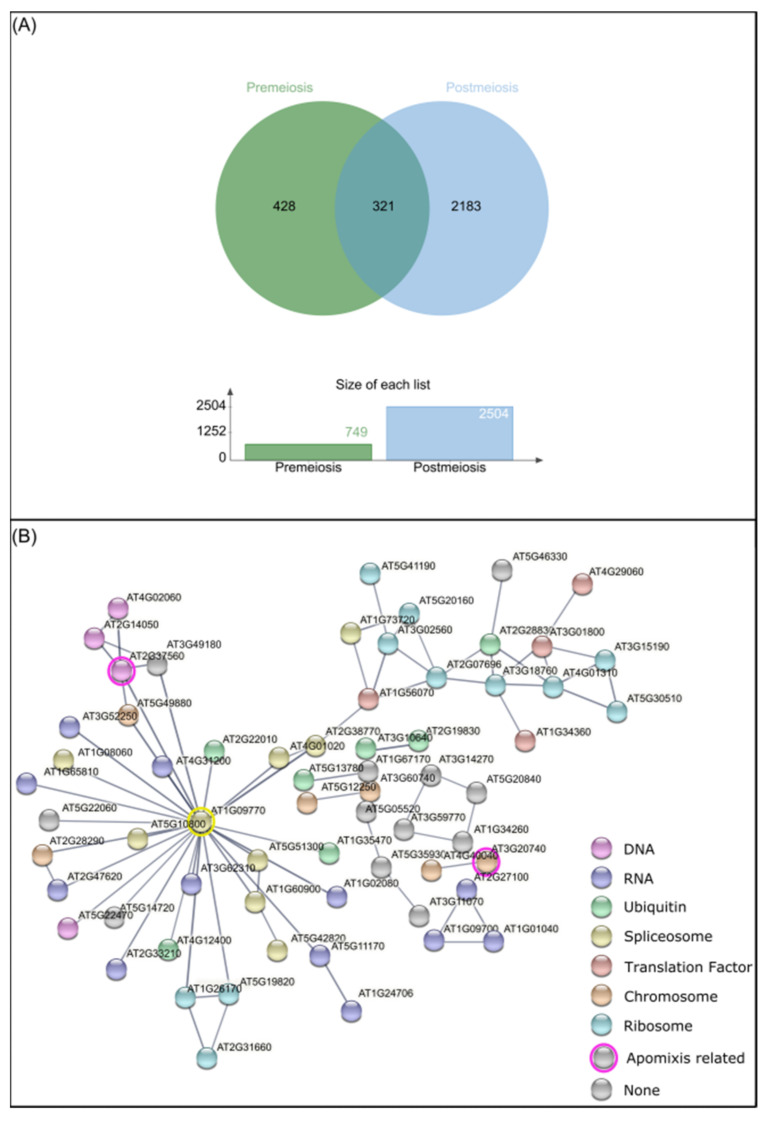
DMCs analysis concerning developmental stages. (**A**) Venn diagram of *Arabidopsis* orthologous genes detected from DMCs at premeiosis (green) and postmeiosis stages (light blue). (**B**) String network between premeiosis specific *A. thaliana* orthologous genes (nodes), connected by known interactions (edges). Different colors represent gene function. **Pink:** DNA replication. **Orange:** chromosome and associated proteins. **Lilac:** RNA degradation, RNA transport, RNA polymerase, transcription machinery, posttranscriptional gene silencing, RNA processing and splicing. **Yellow:** spliceosome. **Light blue**: ribosome. **Red**: translation factors. **Green**: ubiquitin-mediated proteolysis, ubiquitin system and membrane trafficking.

**Figure 6 plants-10-00793-f006:**
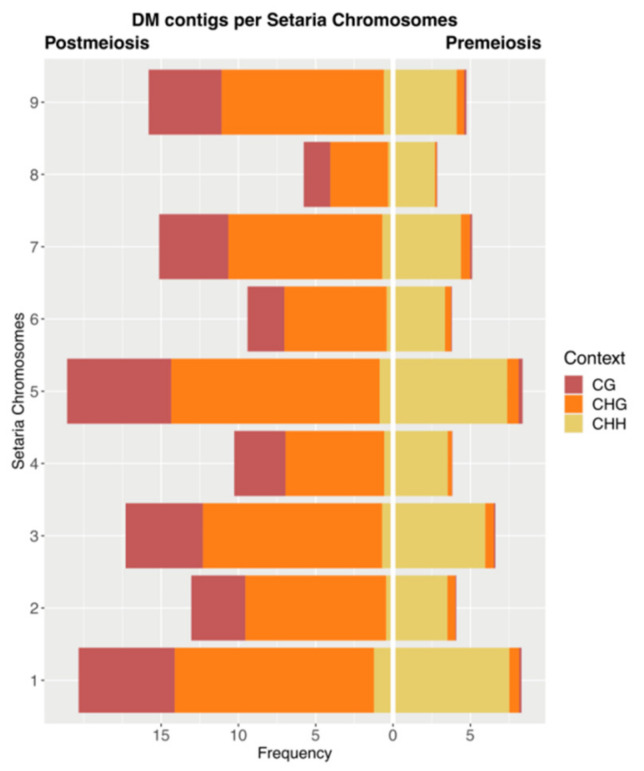
Frequency distribution of DCMs on *Setaria italica* chromosomes at both stages of development, premeiosis (right) and postmeiosis (left) in each methylation context.

**Table 1 plants-10-00793-t001:** Methylation sequencing statistics describing the number of contigs obtained from pseudo-reference genome mapping and quantitative methylation analysis between highly sexual and highly aposporic samples.

Stage	Context	Total Aligned (%) ^1^	DMC ^2^ (%) ^1^	Hypo (%) ^3^	Hyper (%) ^3^
**Premeiosis**	CG	669,290 (56.55)	358 (2.02)	219 (61.2)	139 (38.8)
	CHG	179,662 (15.18)	826 (4.63)	267 (32.3)	559 (67.7)
	CHH	334,594 (28.27)	16,638 (93.36)	9709 (58.4)	6929 (41.6)
**Total**		1,183,546	17,822	10,195	7627
**Postmeiosis**	CG	588,468 (53.87)	11,855 (37.49)	4921 (41.5)	6934 (58.5)
	CHG	189,434 (17.34)	17,676 (55.90)	9116 (51.6)	8560 (48.4)
	CHH	314,474 (28.79)	2087 (6.60)	1143 (54.8)	944 (45.2)
**Total**		1,092,376	31,618	15,180	16,438

^1^% is calculated over the total number of contigs in each stage. ^2^ DMC: differentially methylated contigs. ^3^ hypomethylation or hypermethylation concerning highly sexual samples.

**Table 2 plants-10-00793-t002:** BLAST searches specifications for DMCs corresponding to different methylation contexts and developmental stages concerning different DNA sequences databases.

Stage	Context	Total DMC	No ID ^1^ (%) ^2^	NCBI ID ^3^ (%)	TAIR ID ^4^ (%)	GTA_Pn ^5^ (%)
**Premeiosis**	CG	358	311 (86.87)	33 (9.22)	22 (6.14)	39 (10.89)
	CHG	826	623 (75.42)	149 (18.04)	116 (14.04)	156 (18.89)
	CHH	16,638	13,905 (83.57)	1652 (9.93)	834 (5.01)	2229 (13.40)
**Postmeiosis**	CG	11,855	10,264 (86.58)	1204 (10.16)	852 (7.19)	1274 (10.75)
	CHG	17,676	13,581 (76.83)	3247 (18.37)	2262 (12.80)	3256 (18.42)
	CHH	2087	1689 (80.93)	235 (11.26)	131 (6.28)	324 (15.52)
**Total**		49,440	40,373 (81.66)	6520 (13.19)	4217 (8.53)	7278 (14.72)

^1^ DMCs that did not show any homology on the databases searched. ^2^ % was determined over the total number of DMCs detected for each context and stage. ^3^ Details of DMCs that showed homology against the NCBI NT databases, ^4^ TAIR databases and ^5^
*P. notatum* global transcript assembly.

**Table 3 plants-10-00793-t003:** Specifications of DMCs BLAST search against differentially expressed *Paspalum notatum* floral transcripts between highly aposporic and highly sexual samples.

Stage	Context	Differential Expressed Transcripts ^1^ (%)
**Premeiosis**	CG	11 (28.2)
	CHG	76 (48.72)
	CHH	1038 (46.57)
**Postmeiosis**	CG	511 (40.11)
	CHG	1340 (41.15)
	CHH	151 (46.60)
**Total**		3127 (42.97)

^1^ % was determined over the total number of DMCs homologous to GTA_pn.

**Table 4 plants-10-00793-t004:** Descriptions of DMCs BLAST search against PGSB repeat element database.

Stage	Context	Total	Hyper ^1^	Hypo ^2^	LTR	LTR/ Gypsy	LTR/ Copia	rRNA	DNA	DNA/ En-Spm
**Pre** **meiosis**	CG	0	0	0	0	0	0	0	0	0
	CHG	2	1	1	0	1	1	0	0	0
	CHH	63	40	23	32	14	14	1	2	0
**Post** **meiosis**	CG	15	2	13	5	1	0	9	0	0
	CHG	21	9	12	8	7	4	1	0	1
	CHH	8	2	6	7	1	0	0	0	0
**Total**		109	54	55	52	23	19	11	2	1

^1^ hypermethylated or ^2^ hypomethylated in relation to highly sexual samples.

## Data Availability

Sequences presented in this study are openly available in PRJNA708155. Other data presented in this study are available in [68] and in Appendix A at Appendix A.

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
