# Peer review of "Differential Epigenetic Marks Are Associated with Apospory Expressivity in Diploid Hybrids of Paspalum rufum"

_plants, 2021, doi:10.3390/plants10040793_

Round 1
Reviewer 1 Report
The manuscript plants-1178265 is a paper describing the origin of apomixis as a still unresolved problem. This approach is somehow original since most previous papers do not consider the genomic control of apomixis in such comparative and detailed way. Manuscript is a well written research paper presenting the current understanding of apomixis. In my opinion, the DNA methylation patterns associated with apomixis development in diploid P. rufum genotypes is a worthy topic of investigation.
Generally, it is well written and informative, concise paper. At some points the authors could add list of abbreviations, which will be helpful in understanding next part of the paper. It would be more transparent. Secondly, the first part ‘Introduction’ is too general but I understand that the authors want to define in the beginning that the paper has a focus on apomixis and introduce the basic knowledge of these phenomenon.
Results obtained are based on methylation content sensitive enzyme double-digest restriction-site-associated DNA. Used comprehensive methods allow to get interesting results and to supply the manuscript with high-quality figures. However, in my opinion the text is too extensive and would benefit from some more consolidation and to focus only on significance of obtained data, used references.
Overall, due to the high importance of the subject, difficulty of genetic studies, the manuscript is acceptable for publication. It’s a huge, hardworking, laborious work and my comments are mainly editorial, therefore, in my opinion the manuscript can be accepted for publication in its current form.
Reviewer 2 Report
The study presents differential methylation patterns of apomictic and sexual plants in Paspalum. Overall the study is well designed and well documented. More documentation is needed in the Material and Methods on the bioinformatic analyses. I have further some recommendations for improving introduction/discussion. At some points caution is warranted with interpretation of results.
- Introduction:
Page 2, 2nd paragraph: the authors should mention the recent studies by Schinkel et al. 2020, Int. J. Molec. Sciences (https://doi.org/10.3390/ijms21093318), Syngelaki et al. 2020 in Frontiers in Plant Sciences (doi: 10.3389/fpls.2020.00435) on methylation patterns in apomictic vs. sexual plants (see also notes to discussion).
Page 3, 2nd paragraph: A recent review by Hojsgaard & Hörandl 2019 in Frontiers Plant Sciences summarizes records of apomixis occurrence in diploid plants from various families; in this review there are also quite good arguments that polyploidy only indirectly enhances apomixis frequencies. the idea that polyploidy would be essential for apomixis expression is simply outdated.
The authors could also highlight the value of their study by using diploids only, and hence avoiding well-known ploidy-specific effects on methylation patterns.
Results: section 2.3, Table 2: The high proportions of No ID’s suggest that most of the DMCs are not located in coding regions (and hence are not found in transcriptomes) as typical for these restriction enzymes. Can this be confirmed to exclude other possibilities for the low homology? Should be then considered for discussion/conclusions.
- Discussion:
- 14, 2nd paragraph: “whole-genome methylation patterns” is exaggerated, as the restriction-enzyme approach reveals just a part of the total methylations of the genome (where the enzymes cut). Genome-wide methylations would require a bisulfite-sequencing approach. This restriction should be addressed, and conclusions softened.
- 14, 3rd paragraph: The clustering according to reproductive behaviour is clear. What is not addressed here is the separation of samples within the reproductive groups (ie., in Fig. 1 along the y-axis- observed in sexuals in B, D, and F, and in aposporic ones in C, D, E, and F; also apparent in Fig. 2). There is some mentioning of this variation on p. 15, 1st paragraph, but without any explanation.
Possible reasons: 1) strong genotypic differences between samples? For apomicts it should be also considered that apomixis can fix the methylation pattern over generations much better than sexual plants (because meiotic resetting is avoided) and result in epigenetic divergence (see discussion in Schinkel et al. 2020).
2) Environmental influence? See Schinkel et al. 2020 and Syngelaki et al. 2020 (and literature therein) for environmental effects on methylations. Were the plants kept under controlled and equal conditions before sampling? Were different ecotypes involved?
For this last possible effect it should be kept in mind that also a lot of sporophytic (vegetative) tissue was included, as the authors used whole spikelets (ie., tissue exposed to environment) and not tissue-specific material (ovules).
- 15, 2nd paragraph: here premeiosis is discussed, but the postmeiotic stage not much addressed. In postmeiotic stage difference between sexual /aposporic development could be also due to comparison of reduced/unreduced embryo sacs (see Sharbel et al. 2009, 2010 papers in Boechera on these effects).
- 16, 2nd paragraph: As mentioned above, most DMCs are probably not in coding regions, and hence would not be found in gene bodies. But they still could have influence on gene expression, e.g. if they are located in promoter regions.
And, since DMCs were not complete for coding regions (as it was not whole epigenome-approach), also some genes will be missed if they had no restriction-cutting site. This should be also regarded in the discussion/conclusion.
- Material and Methods:
Vegetal material sounds funny, better “Plant material”
How were the plants kept before sampling to avoid environmental influence on methylation patterns? Experimental garden? Climate growth chambers? What were the conditions? Do plants represent different ecotypes?
- 18, last paragraph ff.: Please describe quality filtering of raw reads, quality measures of your reads (coverage etc.), contig filtering and mapping. The information on p. 4 (1st paragraph) is insufficient for this.
It is also unclear how you discriminated between CG, CHG and CHH contigs. Please describe the analysis with sufficient details so that the study can be replicated by everyone.
- Conclusions: it is no “global” epigenetic analysis. Maybe reword as “epigenetic profile”
Despite the limitations of the study I regard this a valuable contribution to apomixis research.
Minor points:
- 1, 2nd line from bottom: … when it is independent
- 2, 3rd line from top: …. angiosperms it is not ….
- 2, 3rd paragraph, bottom line: correct pseudogamous
- 8, Figure 3: within Figure legend should read “Species”
- 9, header of 2.5.: better Detection of differentially methylated genes related to apomixis
- 10, line space before section 2.6.
- 11, Figure 4: What does Dem mean? I could not find an explanation for this abbreviation.
The text inside the Figure is hard to read (very small). Perhaps a larger font or full page presentation of the Figure would help.
- 14, 2nd paragraph, 2nd line: … approach, which analyses ….
- 15, 2nd paragraph, 3rd line: … sporogenesis occurs ….
